# Fusobacterium nucleatum Infection Induces Malignant Proliferation of Esophageal Squamous Cell Carcinoma Cell by Putrescine Production

Ning Ding,[a] Yikun Cheng,[b] Huijuan Liu,[c] Yueguang Wu,[a] Yongjia Weng,[c] Heyang Cui,[a] Chen Cheng,[a] Weimin Zhang,[a,d] Yongping Cui[a,c]

[a]Cancer Institute, Shenzhen Key Laboratory of Gastrointestinal Cancer Translational Research, Peking University Shenzhen Hospital, Shenzhen Peking University–Hong Kong University of Science and Technology (PKU-HKUST) Medical Center, Institute of Cancer Research, Shenzhen Bay Laboratory, Shenzhen, People's Republic of China
[b]College of Letters & Science, University of California Berkeley, Berkeley, California, USA
[c]Key Laboratory of Cellular Physiology of the Ministry of Education, Department of Pathology, Shanxi Medical University, Taiyuan, Shanxi, People's Republic of China
[d]Key Laboratory of Carcinogenesis and Translational Research (Ministry of Education), Department of Molecular Oncology, Peking University Cancer Hospital and Institute, Beijing, People's Republic of China

Ning Ding and Yikun Cheng contributed equally to this work. Author order was determined in order of decreasing seniority.

**ABSTRACT** Esophageal squamous cell carcinoma (ESCC) is a malignant upper digestive tract cancer, and its pathogenesis and etiology are poorly understood. Because gut microbes commonly impact progression, metastasis, and immunotherapy responses in colorectal cancer (CRC), the roles of the esophageal microbiota in ESCC have gradually drawn attention. As reported previously, Fusobacterium nucleatum (Fn), the notable "culprit" of CRC, can also influence the prognosis of ESCC in clinical studies. However, thus far, the underlying mechanism is unclear. In this study, 73 Chinese ESCC samples were collected. In those clinical samples, the abundance of Fn was found to be higher in tumors than in adjacent normal tissues, and a high abundance of Fn was correlated with shorter survival. Furthermore, using in vitro experiments, we demonstrated that Fn can invade ESCC cells, enhancing their proliferation capacity. The mechanism study revealed that Fn can produce high levels of putrescine after invasion, which disturbs polyamine metabolism and promotes the malignant proliferation of ESCC cells. In conclusion, Fn infection was found in Chinese ESCC tumor tissue samples and may promote ESCC progression by disturbing the polyamine metabolism pathway.

**IMPORTANCE** Nowadays, the complex and varied interactions between microbes and human body are known to be crucial for maintaining the health of the human body. However, knowledge concerning the influence of esophageal microbes on the progression of esophageal squamous cell carcinoma is limited. Here, in our study, we confirmed that F. nucleatum can invade ESCC cells and consequently promote their proliferation, suggesting that esophageal microbes likely influence the progression of ESCC in clinical settings. Because the esophagus connects the oral cavity and stomach, acting as a canal for transporting foods, its special physical location makes it easily exposed to microorganisms. Thus, it is necessary to explore the interaction between esophageal microbes and ESCC.

**KEYWORDS** ESCC, esophageal microbe, Fusobacterium nucleatum, cell proliferation, putrescine

Address correspondence to Yongping Cui, cuiyp@sphmc.org, or Weimin Zhang, zhangweimin@bjmu.edu.cn.

The authors declare no conflict of interest.

Esophageal cancer (EC) is the seventh most diagnosed and the sixth leading cause of cancer deaths worldwide (1). More than 252,500 new cases and 193,900 deaths occur annually in China (2). Most of these cases are diagnosed as esophageal squamous cell carcinoma (ESCC), one of the subtypes of EC (3). Although alcohol consumption, cigarette

smoking, poor oral hygiene, the consumption of hot foods and pickled vegetables, and genetic mutation have been associated with increased risk of ESCC (4–6), its underlying etiology and pathology are still unclear.

Billions of bacteria coexist in/on the human body (7). Among these, the gut microbiota has been the most thoroughly studied. Robust studies have proven the important roles of the gut microbiota in cancer development (8), progression (9, 10), metastasis (11–13), and even immunotherapy responses (14, 15). Like the gut, the esophagus connects the oral cavity and stomach, acting as a canal for transporting food. Its special physical location makes it easily exposed to microorganisms. Indeed, diverse bacteria can be detected in clinical esophageal samples by 16S rRNA gene sequencing or shotgun sequencing, including *Proteobacteria*, *Firmicutes*, *Bacteroidetes*, *Actinobacteria*, and *Fusobacterium* (16, 17). However, little was known about the impact of these local bacteria on ESCC progression.

*Fusobacterium nucleatum* (Fn) is one of the Gram-negative anaerobic bacteria commonly detected in the dental plaque and intestinal tract of the human body (18). Fn is frequently considered an etiological agent of colorectal cancer (CRC) via multiple functional pathways (such as secreting bioactive metabolites, inducing DNA damage, and disturbing the immune system) (14, 19–21). As demonstrated previously, Fn can promote chemoresistance in CRC by targeting TLR4 and MYD88 innate immune signaling and downregulating miR-18a* and miR-4802, subsequently activating autophagy (22). Fn can also target lncRNA EN01-IT1 to promote glycolysis and oncogenesis in CRC (19). As for ESCC, clinical studies have reported that Fn can also be detected in clinical esophageal samples (23–26). These Fn-positive patients exhibit shorter survival and poorer responses to neoadjuvant chemotherapy, but the underlying mechanisms are still unclear.

In the present study, we collected 73 ESCC clinical samples from Shanxi Province, China. We clinically demonstrated that not only was Fn more prevalent in tumor tissues, but it was also significantly correlated with worse prognosis. An *in vivo* study further confirmed that Fn can penetrate ESCC cells and promote malignant proliferation by producing high levels of putrescine and disturbing the polyamine metabolism of ESCC cells after invasion.

## RESULTS

***F. nucleatum* was more abundant in ESCC tumor tissues.** Although it has been demonstrated that *Fusobacterium nucleatum* (Fn) can be detected in ESCC clinical samples and it was correlated with shorter survival in a Japanese cohort (23), whether Fn could be detected in a Chinese cohort and the clinical signature of Fn-positive patients were still unclear. To determine these issues, we detected the relative amount of Fn by qPCR assay in 73 ESCC clinical samples (with paired tumor [T] and adjacent normal tissue [N]) from Shanxi Province, China. Among the 73 clinical samples with paired N&T samples, the relative amount of Fn was significantly higher in tumor tissue than in adjacent normal tissue (Fig. 1A). These 73 Fn-positive samples were further divided into Fn-High and Fn-Low groups according to the relative Fn abundance in tumors. The Fn-High group exhibited significantly shorter survival than the Fn-Low group (Fig. 2B). In conclusion, Fn was significantly elevated in the tumor tissue of Chinese ESCC patients, and this increased Fn infection might have influenced the prognosis of ESCC patients.

**Fn infection induces ESCC cell malignant proliferation and migration.** Because Fn could be detected in ESCC clinical samples and was correlated with shorter survival, we initially intended to determine the influence of Fn infection on the progression of ESCC patients using a Fn and ESCC cell coculture system. The ESCC cell line KYSE70 was incubated with Fn (ATCC 25586) at 400 multiplicities of infection (MOI) for 24 h, and the proliferation and migration ability were analyzed. As shown by CCK8 assay, Fn infection effectively promoted cell viability, whereas heat-killed Fn and another anaerobe, *Bacteroides acidifaciens* (Ba), had little effect on ESCC cell proliferation (Fig. 2A). Similarly, the cell colony-formation experiments also revealed that Fn infection significantly improved the colony formation ability of KYSE70 (Fig. 2B). In addition, using the live-cell imaging system, it was obvious that KYSE70 cells proliferate faster after Fn

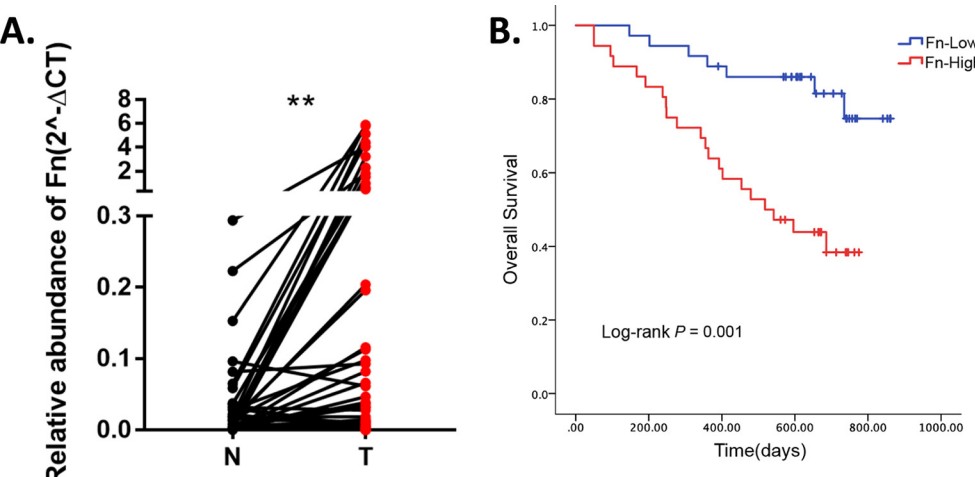

**FIG 1** The relative abundance of *Fusobacterium nucleatum* (Fn) was higher in tumor tissues than in adjacent normal tissues and was correlated with worse prognoses in clinical esophageal squamous cell carcinoma (ESCC) samples from China. (A) Relative abundance of Fn in clinical ESCC samples was detected by TaqMan qPCR analysis. (B) High abundance of Fn infiltration in tumor tissues was significantly correlated with shorter survival. **, $P < 0.01$.

infection compared to cells without Fn infection (Movies S1 and S2). As in the transmission electron microscopy (TEM) results, Fn at the bottom of the petri dish disappeared along with the movement and division of KYSE70 cells (Movie S2), indicating that Fn invaded/adhered/was swallowed into KYSE70 cells. Moreover, the migration ability, another indicator of cancer cell malignance, was also significantly enhanced by Fn infection (Fig. 2C). In conclusion, Fn can promote ESCC cell proliferation and migration.

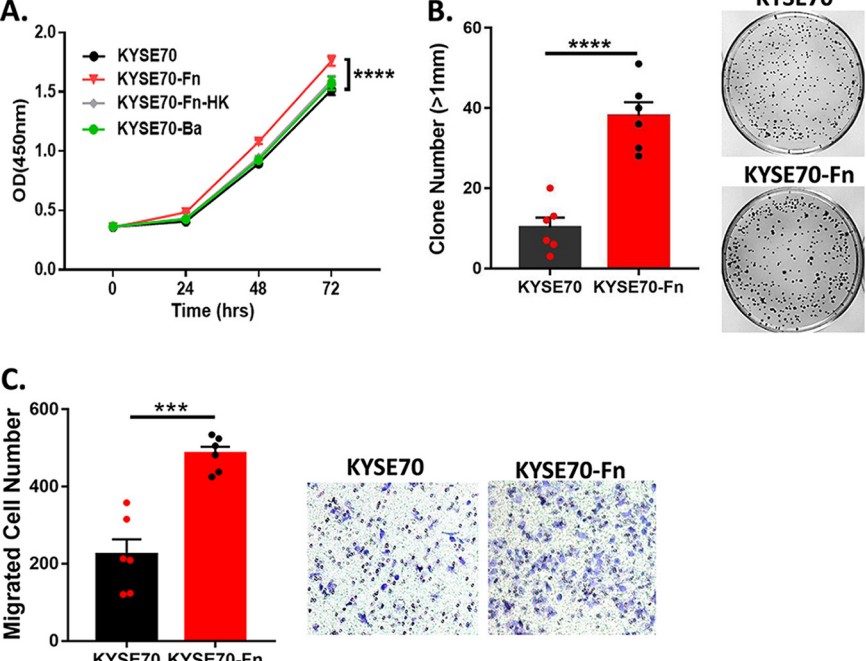

**FIG 2** Fn infection can promote the aggressive proliferation and migration of ESCC cells. (A) Viability of KYSE70 cells with/without Fn infection was examined by CCK8 reagent; heat-killed (HK) Fn and *Bacteroides acidifaciens* (Ba) were used as negative controls. (B) Cell clone formation over 11 days of KYSE70 with/without Fn infection. (C) Migration ability of KYSE70 with/without Fn infection was examined by transwell experiment. The migrated cells from 3 independent experiments were counted and analyzed. ***, $P < 0.001$; ****, $P < 0.0001$.

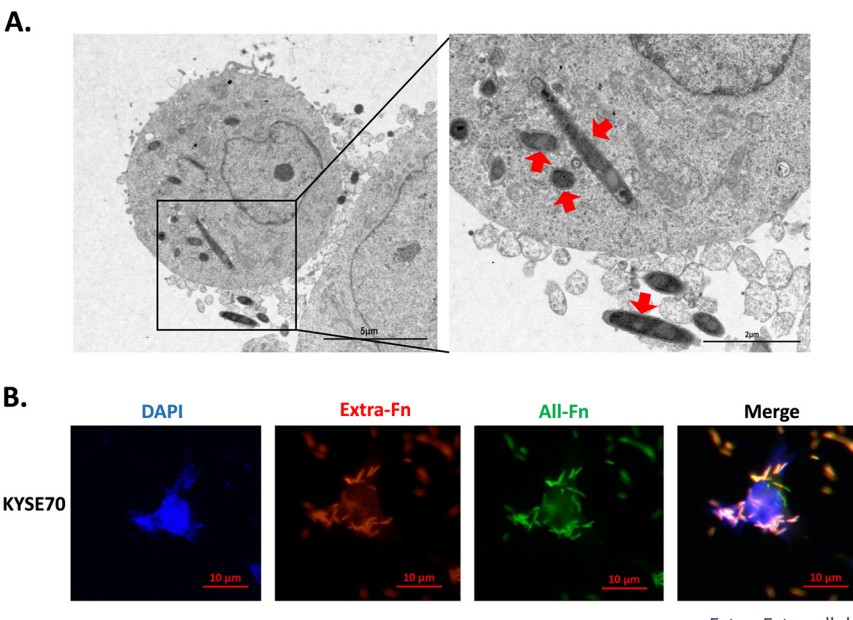

**FIG 3** *F. nucleatum* invasion of KYSE70 cells. (A) Fn-infected cells were detected by transmission electron microscopy. Scale bars = 5 $\mu$m (left), 2 $\mu$m (right). Red arrows indicate *F. nucleatum*. (B) Intracellular Fn was detected by a differential immunofluorescence experiment. Red, Fn outside ESCC cells; green, Fn both inside and outside ESCC cells. Scale bar = 10 $\mu$m.

**Fn can invade ESCC cells.** Because Fn can promote the proliferation of ESCC cells, we further determined whether Fn can invade living ESCC cells. Fn and KYSE70, one of the ESCC cell lines, were cocultured at 400 MOI for 24 h, and then cells were collected for TEM analysis or differential immunofluorescence experiments. In the TEM results, spindle-shaped Fn were found in the cytoplasm of KYSE70 cells (Fig. 3A). As in the TEM results, in the differential immunofluorescence experiment, some intracellular Fn, labeled with green fluorescence in the merged picture, could be detected (Fig. 3B), indicating that Fn can penetrate ESCC cells.

**Fn promotes ESCC cell proliferation by putrescine production which can disturb polyamine metabolism.** To further delineate the potential mechanisms underlying Fn infection-induced malignant cell proliferation, KYSE70 and KYSE70-Fn were collected simultaneously for RNA-sequencing. Spermidine/spermine N1-acetyltransferase 1 (SAT1) was increased significantly after Fn invasion (Fig. 4A). Meanwhile, the relative expression level of SAT1 increased gradually over time after Fn invasion (Fig. 4B). SAT1 encodes a rate-limiting enzyme in the catabolic pathway of polyamine metabolism, predominantly acetylating the $N^1$ position of spermidine and spermine (27). We speculated that the polyamine metabolism pathway was disturbed during Fn infection. Therefore, we determined the polyamine concentrations in ESCC cells with and without Fn infection. Except for spermine, the arginine, ornithine, putrescine, spermidine, $N^1$-acetylspermidine, and $N^1$-acetylspermine concentrations significantly increased (Fig. 4D). Most noteworthy is that the putrescine concentration increased sharply, by more than 6-fold, after Fn infection, indicating that putrescine may play a crucial role in the Fn infection-induced disturbed polyamine metabolism. In addition, by analyzing polyamine concentrations in the Fn microbes themselves, we found that Fn can produce much higher levels of putrescine than other polyamines (Fig. 4E; putrescine, 16.66 nmol/billion Fn; arginine, 6.24 nmol/billion Fn; ornithine, 0.16 nmol/billion Fn; spermidine, 0.34 nmol/billion Fn; spermine, 0.0016 nmol/billion Fn). Meanwhile, the expression level of ornithine decarboxylase (ODC) stabilized in 6 h after Fn infection and decreased significantly at 24 h after Fn infection (Fig. 4F). The ODC gene encodes an enzyme involved in putrescine synthesis, and the decreased ODC expression level indicated that putrescine synthesis in ESCC cells was not active after Fn infection. Thus, the dramatically increased putrescine levels in Fn-infected ESCC cells were produced by Fn.

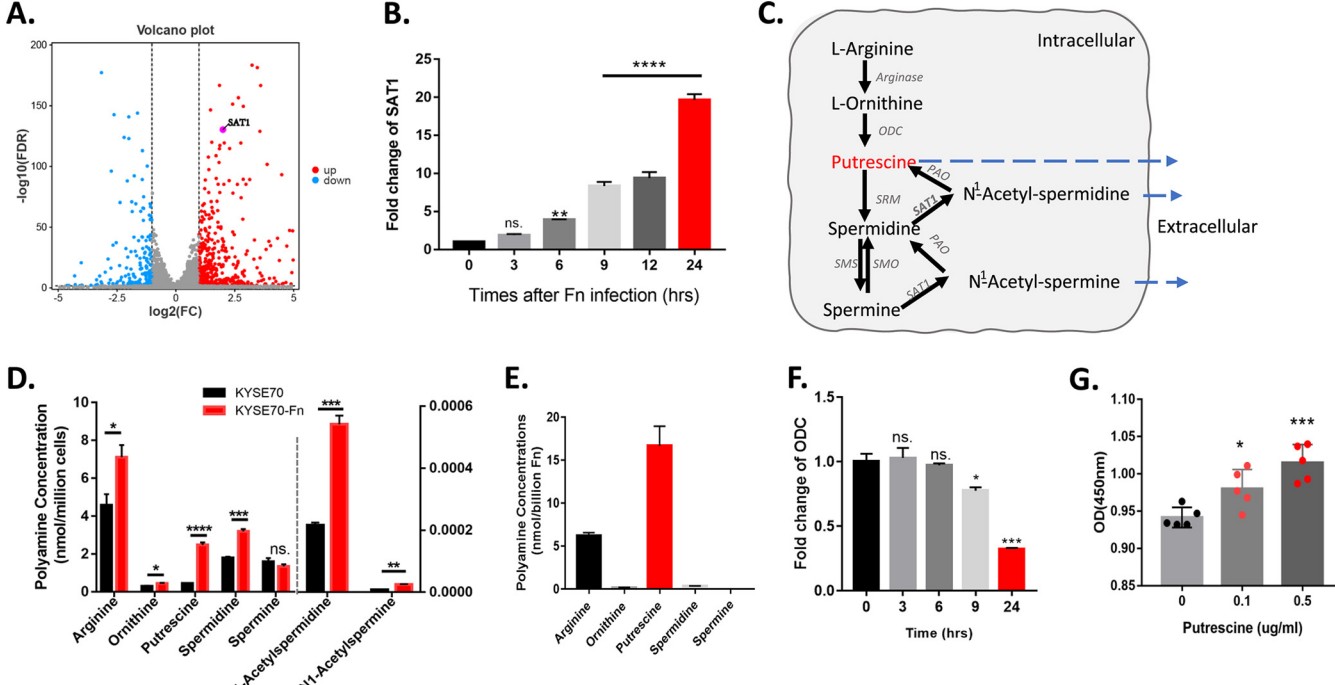

**FIG 4** Fn can produce putrescine, which disturbs the polyamine metabolism pathway in ESCC cells and promotes ESCC cell proliferation. (A) Volcano plot showing the changes in gene expression (fold change of ≥2) in KYSE70 after Fn infection. SAT1 expression increased significantly after Fn infection. (B) Relative expression level of SAT1 increased after Fn infection with time extension. (C) Polyamine metabolism map. Ornithine decarboxylase (ODC) is a rate-limiting step in putrescine production. Spermidine/spermine N1-acetyltransferase 1 (SAT1) can acetylate the N1 position of spermidine or spermine, allowing either export or oxidative back-conversion by peroxisomal acetylpolyamine oxidase (PAO). (D) Polyamine concentrations in ESCC cells with and without Fn infection over 24 h. (E) Polyamine concentration in Fn bacterium. (F) Relative expression level of ODC was stable during the 6 h after Fn infection and decreased sharply at 24 h post-Fn infection. (G) CCK8 assay was used to examine the cell viability of KYSE70 by increased putrescine concentrations. *, $P < 0.05$; **, $P < 0.01$; ***, $P < 0.001$; ****, $P < 0.0001$.

To clarify the influence of elevated putrescine induced by Fn infection, we mimicked the increased cell putrescine level by adding putrescine into the cell culture medium and found that putrescine can increase cell proliferation in a dose-dependent manner (Fig. 4G). In conclusion, the sharply increased putrescine levels in ESCC cells were produced by Fn and can promote the proliferation of ESCC cells.

## DISCUSSION

Esophageal cancer is a common upper gastrointestinal malignancy worldwide; more than half of all cases occur in China, and over 95% were diagnosed as ESCC, one of the histological subtypes of EC. Although numerous genetic abnormalities and gene involvement have been demonstrated (6, 28–30), the underlying etiology and pathology are still unclear. Interestingly, ESCC cases mainly occur in Chaoshan District, the Hazaku community of Xinjiang Province, and Taihang Mountain, and the risk of a person with a positive family history may increase by 2-fold (3), reflecting the regional and family aggregation characteristics of ESCC in China. The microbiome composition is distinct from region to region (31) and is more likely to spread between family members, suggesting that microbes play a role in the occurrence and development of ESCC.

Numerous and varied microbiota exist in/on the human body, with complicated physiological effects, and they can form different ecological communities in response to different lifestyles, locations, and host health conditions (7, 32, 33). The gut microbiota, the most well-known microbiota ecosystem in the human body, can play roles in the development (8), progression (9, 10), metastasis (11–13), and immune therapy (14) of CRC and even some distal carcinomas (15, 34). In addition, tumor tissues which used to be considered sterile (such as breast cancer and pancreatic cancer tissues) have also been colonized by bacteria (13, 35). The esophagus connects the oral cavity and stomach, acting as a

canal for transporting foods. Its special physical location makes it easily exposed to microorganisms (17). We hypothesized that some members of the microbiota can interact with the tumor microenvironment and disturb the progression of ESCC.

*F. nucleatum* is a Gram-negative anerobic bacterium which primarily inhabits the oral cavity. Some clinical studies have demonstrated that esophageal cancer tissues possess more Fn than matched normal tissue and that Fn infection was correlated with shorter survival (23, 24), but the underlying mechanisms were unclear.

Here, in this study, we collected 73 pairs of ESCC tumor (T) and adjacent normal (N) tissue samples form Shanxi Province, China. Clinically, we found that Fn was more abundant in ESCC tumor tissue and was significantly correlated with shorter survival, indicating that Fn infection is a crucial factor that cannot be ignored in the development of ESCC.

The following *in vivo* experiment demonstrated that Fn can invade ESCC cells and subsequently enhance their proliferation ability. The RNA-sequencing and real-time PCR results showed that SAT1, which encodes a rate-limiting enzyme in the catabolic pathway of polyamine metabolism, increased after Fn infection, indicating that polyamine pathway metabolism might be disturbed by Fn infection. Concordantly, the concentrations of putrescine, spermidine, $N^1$-acetylspermidine, and $N^1$-acetylspermine, etc., were indeed increased by Fn infection, and among these, putrescine was the most sharply elevated by more than 6-fold. The subsequent putrescine supplementation experiment demonstrated that elevated putrescine can promote the proliferation of ESCC cells because an increased polyamine pool is important in the proliferation of cancer cells, and the inhibition of both endogenous and exogenous polyamines can reduce the growth rate of lung cancer by more than 80% (36). Conversely, the overproduction of ornithine decarboxylase (ODC) can promote tumor cell malignancy with elevated polyamines (37). Moreover, some inhibitors of polyamine metabolic enzymes, such as difluoromethylornithine (DFMO), have been determined to intervene in cancer progression (38). All of this indicated that polyamine metabolism is quite crucial in tumor progression. The polyamine metabolism pathway disturbed by Fn infection is an important factor which may impact ESCC cell proliferation and the progression of ESCC patients.

Because gut microbes can produce polyamines (27), we analyzed the concentration of polyamines in Fn itself and found that Fn can produce rather high levels of putrescine. Meanwhile, the expression level of ODC, which encodes an enzyme involved in putrescine synthesis, stabilized at 6 h after Fn infection and decreased significantly at 24 h after Fn infection. This indicates that the sharply elevated putrescine levels in Fn-infected ESCC cells were not produced by the ESCC cells, but by Fn.

In summary, in the present study, we demonstrated that Fn was more abundant in Chinese ESCC tumor tissues and was significantly corelated with shorter survival. Using an *in vitro* experiment, we found that Fn can invade ESCC cells and promote their proliferation by elevating the concentration of polyamine metabolites. In particular, Fn can produce high levels of putrescine, inducing the malignant proliferation of ESCC cells.

The fact that Fn can invade ESCC cells presents a meaningful question: how does Fn invade these cells? Some studies have demonstrated that Fn can bind to an 11-amino acid region of E-cadherin by its FadA virulence factor and induce CRC cell growth (39). In addition, the Fap2 of Fn can also bind to the D-galactose-$\beta$(1,3)-*N*-acetyl-D-galactosamine (Gal-GalNAc) of cancer cells in CRC and breast tumor cells (40, 41). However, it is still unclear which factor promotes bacterial entry into these cells, as it has been demonstrated that *Listeria monocytogenes*, a pathogenic bacterium, can invade HeLa cells by clathrin-dependent endocytosis (42), and that enteropathogenic *Escherichia coli* can hijack a conserved set of endocytic proteins to promote its internalization (43). Thus, it would be interesting in further studies to determine whether Fn can enter cells by endocytosis.

Several limitations should be noted here. First, only cell experiments were carried out in the present study. Because the tumor microenvironment is a complicated system, the effect of *F. nucleatum* infection should be studied more thoroughly by an *in*

*vivo* study. Second, because the immune system plays an important role in tumor formation, progression, and therapy, the interaction between Fn and the immune system should be considered.

## MATERIALS AND METHODS

**Patients.** A total of 73 patients form Shanxi Province diagnosed with ESCC who had received no prior treatment were recruited in this study. Informed consent was obtained from all subjects. This study was approved by the ethical committees of Shenzhen Bay Laboratory.

**Fn detection by TaqMan qPCR.** Total gDNA from Fn DNA in clinical ESCC samples was extracted and the Fn DNA in those clinical samples of gDNA was subsequently detected by TaqMan qPCR, as previously described (23). Briefly, the amplifications were performed by Universal U+ probe Master Mix V2 (Vazyme Biotech, Q513-02) following the manufacturer's instructions. The following primer and probe sequences were used: Fn forward primer, 5′-TGGTGTCATTCTTCCAAAAATATCA′; Fn reverse primer, 5′-AGATCAAGAAGGACAAGTTGCTGAA′; Fn FAM probe, 5′-ACTTTAACTCTACCATGTTCA′; PGT forward primer, 5′-ATCCCCAAAGCACCTGGTTT′; PGT reverse primer, 5′-AGAGGCCAAGATAGTCCTGGTAA′; PGT FAM probe, 5′-CCATCCATGTCCTCATCTC′. Amplification and detection was performed with the ABI StepOne Plus real-time PCR system under the following reaction conditions: 5 min at 95℃, followed by 40 cycles of denaturation at 95℃ for 10 s and at 60℃ for 30 s. The cycle threshold ($C_T$) values of Fn were normalized to the $C_T$ value of PGT (44).

**Cell culture.** Human ESCC cell line KYSE70 was cultured in RPMI 1640 medium supplemented with 10% fetal bovine serum (FBS) and 1% penicillin-streptomycin solution (PS). Cells were incubated at 37℃ and 5% $CO_2$.

**Bacterium culture.** *F. nucleatum* (ATCC 25586) was obtained from the Guangdong Microbial Culture Collection Center (GDMCC) and incubated in tryptic soy broth at 37℃ under anaerobic conditions. *B. acidifaciens* (DSMZ 15896) was obtained from the DSMZ-German Collection of Microorganisms and Cell Cultures GmbH and incubated in brain-heart infusion broth at 37℃ under anaerobic conditions.

**Bacterium invasion.** KYSE70 ESCC cells were seeded in a 96-well polypropylene microplates with complete cell culture medium (1640 medium with 10% FBS and 1% PS). Before Fn invasion, cells were washed twice with phosphate-buffered saline (PBS) solution and antibiotic-free medium was added. Fn was collected during the logarithmic phase, washed twice with PBS, and added to the cells at a MOI of 400, and cells were incubated in the cell incubator for 24 h. After infection, cells were washed with PBS, fresh complete medium was added, and cells were incubated for the following experiments.

**Differential immunofluorescence experiment of Fn.** Intracellular bacteria detection has been described previously (45). Briefly, the ESCC cells with/without Fn infection for 24 h were fixed with 4% paraformaldehyde fix solution for 15 min, and blocked by 5% donkey serum (Jackson, 017-000-121) for 1 h at room temperature. Next, the extracellular Fn were reacted with primary antibody (Diatheva, ANT0084) at 1:200 at for 1 h at room temperature; after PBS washes, the primary antibody was revealed by the secondary antibody Alexa-594 (Thermo Fisher Scientific, cat no. A32753) for 1 h. After PBS washes, cells were permeabilized for 20 min with 0.1% Triton X-100. A second round of primary antibody reaction were reacted as previously: after PBS washes, the primary antibody was revealed by secondary antibody Alexa-488 (Thermo Fisher Scientific, cat no. A32790) for 1 h. After washing, the coverslips were mounted onto slides with ProLong Gold Antifade Mountant (Thermo Fisher Scientific, cat no. P36935).

**Cell proliferation detection.** The proliferation of ESCC cell lines was measured by CCK8 assay. Briefly, ESCC cells with or without Fn invasion were washed with PBS before detection. Next, 100 $\mu$L of fresh cell culture medium containing 10% CCK8 solution was added, and cells were kept in the incubator for 1 h. Cell viability was then determined by measuring the absorbance at a wavelength of 450 nm.

**Colony formation assay.** For the colony formation assay, 600 cells were seeded in a 60-mm petri dish containing complete RPMI 1640 on day 1. The medium was replaced with PS-free cell culture medium on day 2, then Fn or PBS was added into cell plate at 400 MOI and cells were kept in the incubator for 24 h. The medium was replaced with complete RPMI 1640 medium on day 3 and maintained for another 8 days. On day 11, cells were fixed with methanol for 10 min and stained with 1% crystal violet. After washing with running water, cell colonies with a diameter of more than 1 mm were counted.

**Cell migration detection.** The migration ability of ESCC cells with/without Fn infection was measured by transwell assay. Briefly, $2 \times 10^5$ ESCC cells with or without Fn infection for 6 h were resuspended in 2% FBS medium and seeded into the upper chambers of a 6.5 mm $\times$ 8.0 $\mu$m transwell (Corning, cat no. 3422). The bottom chambers contained 600 $\mu$L 20% FBS medium. After 40 h, the transwell chambers were sequentially washed with PBS, fixed with methanol, and stained with 0.1% crystal violet. After staining, the upper chambers were wiped gently with cotton swabs. Microscopy was used to image cells which migrated to the underside of the transwell membrane. Four fields were randomly selected, and the migrated cells were counted from those fields.

**Live-cell imaging.** For live-cell imaging, $2 \times 10^5$ KYSE70 cells were seeded in 6-well plates 24 h in advance, and logarithmic-phase Fn was added to the plates. Next, the coculture system was photographed by a BioTek Cytation 1 for more than 48 h.

**Real-time PCR.** Real-time PCR was performed to measure the expression levels of differentially expressed genes of interest in ESCC cell lines. In brief, the extracted qualified total RNA was reversely transcribed into cDNA with commercialized kits (Vazyme, China), and then the prepared cDNA samples were subjected to real-time PCR using SYBR Green reagent (Vazyme, China) according to the manufacturer's protocol. The primer sequences of targeted genes are listed in Table 1. All reactions were performed in

**TABLE 1** Real-time PCR primer sequences

| Gene | Forward sequence (5′–3′) | Reverse sequence (5′–3′) |
|------|--------------------------|--------------------------|
| SAT1 | TGGTTGCAGAAGTGCCGAAAGAG | CTTGCCAATCCACGGGTCATAGG |
| ODC | CAGAGCACATCCCAAAGCAAAGTTG | TCTGAGCGTGGCACCGAATTTC |
| PGK1 | GAGATGATTATTGGTGGTGGAA | AGTCAACAGGCAAGGTAATC |

triplicate with an Applied Biosystems StepOnePlus system. Data were normalized to PGK1 expression ($\Delta\Delta C_T$ analysis).

**Polyamine detection by UHPLC-MS/MS.** The ESCC cells with/without Fn infection for 24 h were collected in 80% aqueous methanol. After ultrasonic treatment, the lysates were centrifuged, and the supernatant was dried and dissolved in 50 $\mu$L of 100 mM sodium carbonate. The chemical derivatization was initiated after adding 50 $\mu$L of 2% benzoyl chloride in acetonitrile. After derivatization, the sample was isometrically mixed with benzoyl-13C6 chloride-derivatized standards (as internal standards) prior to ultra-high-performance liquid chromatography–high-resolution tandem mass spectrometry (UHPLC-HRMS/MS) analysis. The UHPLC-MS/MS analysis was performed on an Agilent 1290 Infinity II UHPLC system coupled to a 6470A Triple Quadrupole mass spectrometer (Agilent, Santa Clara, United States).

The reagents used to prepare for standard solution, such as L-arginine, L-ornithine, putrescine, spermine, and spermidine, were purchased from Sigma-Aldrich. N1-acetylspermidine and N1-acetylspermine were purchased from Shanghai Yuanye Bio-Technology Co (Shanghai, China).

**Statistical analysis.** Data are expressed as mean $\pm$ standard error of the mean. All experiments (except the live-cell imaging) were repeated three times independently. Statistical comparisons between two groups were analyzed by unpaired $t$ test using GraphPad Prism 7.0. Statistical comparisons conducted in more than two groups were analyzed by two-way analysis of variance. $P < 0.05$ was considered to be statistically significant.

**Data availability.** Data from this study have been deposited in Genome Sequence Archive (Genomics, Proteomics & Bioinformatics 2021) in National Genomics Data Center (Nucleic Acids Res 2022), China National Center for Bioinformation / Beijing Institute of Genomics, Chinese Academy of Sciences (GSA-Human: HRA003868) that are publicly accessible at https://ngdc.cncb.ac.cn/gsa-human.

Other data and sources associated with this study are available from the corresponding author on reasonable request.

## SUPPLEMENTAL MATERIAL

Supplemental material is available online only.

**SUPPLEMENTAL FILE 1**, MPG file, 5.6 MB.
**SUPPLEMENTAL FILE 2**, MPG file, 7.6 MB.

## ACKNOWLEDGMENT

This work was supported by fund from the Guangdong Basic and Applied Basic Research Foundation (2019B030302012), the Major Program of Shenzhen Bay Laboratory (S201101004), the Shenzhen Key Project of Science and Technology (JCYJ20200109120425045), the Shenzhen Bay Laboratory Open Program (SZBL2020090501003), the National Natural Science Foundation of China (U21A20372, 81972613, 82103143), the China National Postdoctoral Program for Innovative Talents (BX2021194), the National Key R&D Program of China (2021YFC2501001), the "San-ming" Project of Medicine in Shenzhen (SZSM201812088), and the Natural Science Foundation of Guangdong Province (2020A1515010431).

N.D., Y.C., H.L., YW., and Y.W. performed experiments and data analysis. Y.C. collected clinical samples and detected Fn abundance in ESCC samples. N.D. wrote the manuscript. C.C. and H.C. revised the manuscript. W.Z. and Y.C. designed and guided this project. All authors reviewed and approved the final manuscript.

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
