## [Reviewer comments · Microbiology Spectrum]

Microbiology Spectrum

***Fusobacterium nucleatum* infection induces malignant proliferation of esophageal squamous cell carcinoma cell by putrescine production**

Yongping Cui, Weimin Zhang, Ning Ding, Huijuan Liu, Yueguang Wu, Yongjia Weng, Heyang Cui, and Chen Cheng

Corresponding Author(s): Yongping Cui, Shenzhen Key Laboratory of Gastrointestinal Cancer Translational Research, Peking University Shenzhen Hospital, Shenzhen Peking University-the Hong Kong University of Science and Technology (PKU-HKUST) Medical Center. Institute of Cancer Research, Shenzhen Bay Laboratory

Review Timeline:

Submission Date:	July 20, 2022
Editorial Decision:	September 3, 2022
Revision Received:	December 16, 2022
Accepted:	January 13, 2023

Editor: Florence Doucet-Populaire

Reviewer(s): Disclosure of reviewer identity is with reference to reviewer comments included in decision letter(s). The following individuals involved in review of your submission have agreed to reveal their identity: OLATUNJI KAYODE AREMU (Reviewer #2); Yanlei Ma (Reviewer #3); Ali Murad Al-Fendi (Reviewer #4)

Transaction Report:

DOI: <https://doi.org/10.1128/spectrum.02759-22>

September 3, 2022

Dr. Yongping Cui

Peking University Shenzhen Hospital, Shenzhen Peking University-the Hong Kong University of Science and Technology (PKU-HKUST) Medical Center;
Cancer Institute, Shenzhen Key Laboratory of Gastrointestinal Cancer Translational Research,
Shenzhen
China

Re: Spectrum02759-22 (Fusobacterium nucleatum infection induces malignant proliferation of esophageal squamous cell carcinoma cells)

Dear Dr. Yongping Cui:

Link Not Available

Sincerely,

Florence Doucet-Populaire

Journals Department
Reviewer comments:

Reviewer #1 (Comments for the Author):

Fusobacterium nucleatum is a cancer-related bacterium. A large number of studies have revealed the relationship between Fn and cancers from different angles since 2012. However, those researches mainly focus on colorectal cancer. For the relationship of Fn and esophageal cancer, the Japanese first reported in Clin Cancer Res in 2016 that high load of Fn was positively correlated with the short survival period of esophageal cancer. In 2019, this team found that Fn reduced the effect of neoadjuvant chemotherapy. In 2022, they observed with transmission electron microscope that Fn could invade ESCC cells and affect gene and protein expression, and found the promotion of NOD1 receptor - RIPK2 -NF- κB in the invasion, migration and

proliferation of cancer cells. In addition, in 2021, through the analysis of 41 Chinese patients with esophageal cancer, Zhen Li et al found that the Fn was significantly related to the pathological stage and clinical stage of esophageal cancer, and also to the metastasis of tumor.

In order to explore the mechanism of Fn on esophageal cancer, the ESCC cell line and Fn were co-cultured in this study. Two conclusions were obtained: 1. Fn can invade ESCC cells, thus enhancing the proliferation of ESCC cells. 2. After Fn infection, gene expression in ESCC cells was changed, and gene expression related to inflammation related signals and cancer signals was up-regulated.

There are few studies on the correlation between Fn and esophageal cancer, and its corresponding mechanism is not clear. The topic of this study is very meaningful, and the language description of the manuscript is clear. However, the mechanism of Fn and the proliferation of esophageal cancer has not been clearly proposed. It is suggested to supplement the corresponding data to better understand the pro-proliferative effect of Fn in esophageal cancer.

The suggested supplementary data are as follows:

1. Why is there a significant difference in the morphology of Fn in Fig. 1? How can you prove that the marker is a bacterium, not a cell structure or other components?
2. In Fig2b, Does *Bacteroides acidifaciens* also enter the cell?
3. In this study, there is only co-culture model in vitro, and corresponding animal experiments should be supplemented to observe whether Fn promotes the proliferation of esophageal cancer in vivo;
4. To clarify the specific target and corresponding mechanism of Fn promoting the proliferation of esophageal cancer;

Reviewer #2 (Comments for the Author):

Most of the information provided in this study were not arranged sequentially. The author needs to follow standard procedure so that the study will be easier to understand.

Please note my input below:

Topic - I am suggesting a new title: Induction of Malignant Proliferation of Esophageal Squamous Cell Carcinoma by *Fusobacterium nucleatum*

Abstract - You need to be focused on your topic. Give a concise summary of the background of the study, objective, methods used for the study, result obtained (findings) and conclusion (in line with the objective). Please use this concept to re-write your abstract. Presently every thing is muddled up.

Introduction - You started well but veered off. Some introduction information were stated under discussion without sufficient references

Method - It will carry more weight and portray a better flow if it comes before discussion, Please relocate.

Result - You veered off. The headings were not concise. Please report what you found out

Reviewer #3 (Comments for the Author):

Through a series of in vitro experiments, Ding and colleagues found that Fn could invade ESCC cells and further enhance the proliferation ability of ESCC cells. In addition, the expression levels of CXCL1, IL-6 and IL-1 β in ESCC cells were significantly increased after Fn infection. I hope the authors may find some of the following suggestions useful:

Major points:

1. There is no conclusion description in the abstract.
2. There is no validation and molecular biological rescue experiments For RNA-seq results regulated by Fn.
3. What is the specific pathway of Fn invasion into cells? It should be discussed or validated accordingly
4. For the change of tumor cell function caused by Fn, it is not enough to only observe the proliferation ability. Does it affect the invasion, migration and apoptosis of cells?
5. The abundance of Fn in clinical patients should be supplemented. Is there a correlation between Fn level and clinicopathological parameters?

Staff Comments:

Preparing Revision Guidelines

Please return the manuscript within 60 days; if you cannot complete the modification within this time period, please contact me. If you do not wish to modify the manuscript and prefer to submit it to another journal, please notify me of your decision immediately so that the manuscript may be formally withdrawn from consideration by Microbiology Spectrum.

***Fusobacterium nucleatum* Infection Induces Malignant Proliferation of Esophageal Squamous Cell Carcinoma Cells**

Ning Ding^a, Huijuan Liu^b, Yueguang Wu^a, Yongjia Weng^b, Heyang Cui^a, Chen Cheng^a, Weimin Zhang^{#a,c}, Yongping Cui^{#a,b}

^a Cancer Institute, Shenzhen Key Laboratory of Gastrointestinal Cancer Translational Research, Peking University Shenzhen Hospital, Shenzhen Peking University-the Hong Kong University of Science and Technology (PKU-HKUST) Medical Center; Institute of Cancer Research, Shenzhen Bay Laboratory, Shenzhen 518028, P. R. China.

^bKey Laboratory of Cellular Physiology of the Ministry of Education, Department of Pathology, Shanxi Medical University, Taiyuan, Shanxi 030001, P. R. China.

^cKey Laboratory of Carcinogenesis and Translational Research (Ministry of Education), Department of Molecular Oncology, Peking University Cancer Hospital and Institute, Beijing 100142, P. R. China.

Running Head: *F.nucleatum* infection enhanced ESCC cell proliferation

#Address correspondence to Yongping Cui, cuiyp@sphmc.org and Weimin Zhang, zhangweimin@bjmu.edu.cn.

Words count for the abstract: 196 words

Words count for the importance:111 words

Words count for the text: 1992 words

Abstract

Esophageal Squamous Cell Carcinoma (ESCC) is a malignant upper digestive tract cancer with poorly understanding of its pathogenesis and etiology. Along with gut microbes can impact the progression, metastasis and immunotherapy response of colorectal cancer (CRC) becoming a common sense, the roles of esophageal microbiota in ESCC drew attention gradually. As reported, *Fusobacterium nucleatum* (Fn), the notable “culprit” of CRC, also can influence the prognosis of ESCC in clinical studies. However, thus far, the underlying mechanism is unclear. To address this issue, ESCC cell lines and Fn were co-cultured, and demonstrated that Fn can invade into ESCC cells, and subsequently enhancing the proliferation capacity of ESCC cells by in vitro experiments. Otherwise, 700 genes were expressed differentially in the KYSE70 cells after Fn infection. Among them there were 493 genes with increased expression levels, and 207 genes with decreased expression levels. Those differentially expressed genes was mainly enriched in inflammation relative signaling pathways and cancer signaling pathways. Indeed, the expression level of CXCL1, IL-6 and IL-1 β in ESCC cells increased dramatically after Fn infection, indicating a pro-inflammation condition. Otherwise, the expression of ESCC progression relative genes, including CCND1, NFE2L2, HAS3, DKK1 were also changed significantly.

Importance

Nowadays, the complex and variety interactions between microbes and human body presenting to be crucial for keeping human body health. However, the knowledge concerning the influence of esophageal microbes on the progression of Esophageal squamous cell carcinoma is limited. Here in our study, we confirmed that the *Fusobacterium nucleatum* can invade into ESCC cells and promoted the cell proliferation consequently. Suggesting the esophageal microbes are likely to influence the progression of ESCC in clinical. As esophageal connects oral cavity and stomach, acts as a canal for transporting foods. Its special physical location makes it easy

exposing to microorganism. Thus, it is necessary to explore the interaction between esophageal microbes and ESCC.

Introduction

Esophageal cancer is the seventh most diagnosed and the sixth leading cause of cancer deaths worldwide(1). More than 252500 new cases and 193900 annual deaths occurring in China(2). Majority of these cases are diagnosed as Esophageal squamous cell carcinoma (ESCC), one of the subtypes of EC (3). Although, alcohol consumption, cigarette smoking, poor oral hygiene, intake hot foods and pickled vegetables, and genetic mutation have been associated with increased risk of ESCC(4-6), the underlying etiology and pathology still not clear.

Billions of bacterium are co-existing in/on human body(7). Among them gut microbiota has been studied most thoroughly. Robust studies provided the important role of gut microbiota in cancer development(8), progression(9, 10), metastasis(11-13), and even immunotherapy response(14, 15). Same as the gut, esophageal connects oral cavity and stomach, acts as a canal for transporting foods. Its special physical location makes it easy exposing to microorganism. Indeed, diverse bacteria can be detected in clinical esophageal samples by 16s-rDNA sequencing or shot-gun sequencing, including *Proteobacteria*, *Firmicutes*, *Bacteroidetes*, *Actinobacteria* and *Fusobacterium*(16, 17). However, little was known about the impacts of those local bacterium on ESCC progression.

Fusobacterium nucleatum (Fn) is one of the Gram-negative anaerobic bacteria commonly detected in the dental plaque and intestinal tract of human body(18). Fn is frequently considered as an etiological agent of colorectal cancer (CRC) via multiple functional pathways (such as secreting bioactive metabolites, inducing DNA damage and disturbing immune system)(14, 19-21). As demonstrated, Fn can promote chemoresistance in CRC by targeting TLR4 and MYD88 innate immune signaling and downregulation miR-18a* and miR-4802 subsequently activation autophagy(22). Fn also can targets lncRNA EN01-IT1 to promote glycolysis and oncogenesis in CRC(19). As for ESCC, clinical studies reported Fn can also be detected in the clinical esophageal samples(23-26). Those Fn positive patients are associated with

shorter survival and poorer response to neoadjuvant chemotherapy, but the underlying mechanism are still not clear.

In the present work, we found that Fn can penetrate and promoted the proliferation of ESCC cells. Along with Fn invading, there were 700 genes expressed differentially, among them there were 493 genes with increased expression levels and 207 genes with decreased expression levels. Those differentially expressed genes (DEGs) were mainly enriched in immunity relative signaling pathways, including the TNF signaling pathway, the IL-17 signaling pathway, the cytokine-cytokine receptor interaction pathway, and the chemokine signaling pathway. Concordantly, the dramatically elevated expression of cytokines were verified by Real-time PCR, such as IL-6, IL-1 β and CXCL1. Additionally, the expression of ESCC progression relative genes, such as CCND1, NFE2L2, HAS3, DKK1 were changed significantly. Taken together, this study demonstrates Fn can invade into ESCC cells and promote the proliferation of ESCC cells along with elevated expression of cytokines (CXCL1, IL-6 and IL-1 β) and disturbed expression of some ESCC progression associated genes (DKK1, HAS3, NFE2L2 and CCND1).

Results

1. Fn can invade ESCC cells by co-culture

Fusobacterium nucleatum (Fn) can be detected in ESCC clinical samples and were correlated with shorter survival(23). Thus, we initially determined whether Fn can invade into living ESCC cells. The Fn (ATCC25586) and KYSE70, one of the ESCC cell lines, were co-cultured at 400 multiplicity of infection (MOI) for 24hrs, and then cells were collected for Transmission electron microscopy (TEM) analysis. The spindle shaped Fn can be found in the cytoplasm of KYSE70 cells (Fig. 1), indicated Fn can penetrate living ESCC cells.

Fig.1 *F.nucleatum* invaded into KYSE70 cells as detected by transmission electron microscopy. Scale bar = 5µm (left), 2µm (right). Red arrows indicate *F.nucleatum*.

2. Fn infection induces ESCC cell malignant proliferation

To further figure out the influence of Fn infection in ESCC cells, KYSE70 and TE-5 cells were incubated with Fn at MOI400 for 24 hrs. As shown by CCK8 assay, Fn infection effectively promoted cell viability, whereas the heat killed Fn and another anaerobe *Bacteroides acidifaciens* (Ba) have little effect on ESCC cell proliferation (Fig. 2A). Similarly, the cell colony formation experiments also revealed that Fn infection significantly improved the colony formation ability in KYSE70 (Fig. 2B). In addition, with the living cell imaging system, it is obvious that KYSE70 cells are proliferating faster after Fn infection compared to cells without Fn infection (Movie

S1, S2). Same as the TEM results, Fn that at the bottom of the petri dish were disappeared along with the movement and division of KYSE70 cells (Movie S2), which indicated that Fn invaded/adhered/swallowed into KYSE70 cells. Taken together, Fn can not only invade into ESCC cells but also promote cell proliferation.

Fig 2. Fn infection can enhance the aggressive proliferation of ESCC cells. A. The cell viability of KYSE70 and TE5 cells with/without Fn infection was examined by CCK8 reagent, the heat killed (HK) Fn and *Bacteroides acidifaciens* (Ba) are used as negative controls. B. Cell clone formation for 11 days of KYSE70 with/without Fn infection. **** p<0.0001.

3 RNA-seq reveals a pro-inflammation signature associated with Fn infection

To further delineate the potential mechanisms underlying Fn infection induced cell proliferation changing, KYSE70 and KYSE70-Fn were collected simultaneously for RNA-sequencing. 700 DEGs were identified with more than two-fold expression change (FDR<0.05). Among them, there were 207 genes with decreased expression levels, and 493 genes with increased expression levels (Fig. 3A). Those DEGs significantly enriched in the pathways related to immunity, such as TNF signaling pathway, IL-17 signaling pathway, cytokine-cytokine receptor interaction pathway

and chemokine signaling pathway by Kyoto Encyclopedia of Genes and Genomes (KEGG) pathway analysis (Fig. 3B). The up-regulation expression levels of pro-inflammatory cytokines IL-1 β , CXCL1 and IL-6 were verified by Real-time PCR (Fig. 3C). Indicating that Fn invasion provoke a pro-inflammation signature in the ESCC cells. Otherwise, the expression levels of ESCC progression relative genes, including CCND1, DKK1(27), NFE2L2, HAS3(28) were also verified by Real-time PCR (Fig. 3D). All these changes might be conducive to the rapid proliferation of ESCC cells.

Fig 3. Alterations of gene expression after Fn invasion ESCC cells. A. Volcano plot presenting the expression level changes of genes in KYSE70 after Fn infection. B. The top 20 pathways was presented by KEGG pathway analysis. C. The mRNA expression levels of IL-6, IL-1 β and CXCL1 in KYSE70 and KYSE70-Fn. D. The mRNA expression levels of ESCC progression relative genes (DKK1, HAS3, NFE2L2 and CCND1) in KYSE70 and KYSE70-Fn. * $p < 0.05$, *** $p < 0.001$, **** $p < 0.0001$.

Discussion

Esophageal cancer (EC) is a common upper gastrointestinal malignancy worldwide, with more than half cases occurs in China and over 95% were diagnosed as ESCC, one of the histological subtype of EC. Although numerous genetic abnormalities and gene involvement were demonstrated(6, 29-31), the underlying etiology and pathology are still not clear. Interestingly, ESCC cases are mainly occurs in Chaoshan district, the hazaku community of Xinjiang province, and Taihang Mountain, and the risk of person who with a positive family history may increase by 2-fold(3), reflecting the regional and family aggregation characteristic of ESCC in China, suggesting microbes might play roles in the occurrence and development of ESCC.

Numerous and variety microbiota are existed in/on human body with complicated physiological effects and can form different ecological communities in response to different lifestyle, location, and health condition of the host(7, 32, 33). Gut microbiota, the most well-known microbiota ecosystem in human body, can play roles in the development(8), progression(9, 10), metastasis(11-13), and immune therapy (14) of CRC and even some distal carcinoma(15, 34). Otherwise, tumor tissues which were used to be considered as sterile (such as breast cancer and pancreatic cancer) were also colonized by bacteria (13, 35). Esophageal connects oral cavity and stomach, acts as a canal for transporting foods. Its special physical location makes it easy exposing to microorganism(17). We supposed some of the microbiota can interact with tumor microenvironment and disturb the ESCC progression.

Fusobacterium nucleatum, the gram-negative anerobic bacterium which are primarily inhabits the oral cavity. Some clinical studies have demonstrated that esophageal cancer tissues possess more Fn than matched normal tissue and were correlated with shorter survival(23, 24), but the underlying mechanism were not clear.

Here in this study, we demonstrated that Fn can invade into KYSE70 cells, one of the ESCC cell lines, and subsequently enhanced the proliferation ability of ESCC cells by a bacterium and ESCC cell co-culture system. Furthermore, the RNA-sequencing and real-time PCR results showed that the ESCC cells presenting a pro-

inflammation signature with differentially expressed genes enriched in TNF and IL-17 signaling pathway and dramatically elevated expression of CXCL1, IL-1 β and IL-6 after Fn infection. As inflammation conditions can increase the risk of cancer, and the cytokines in the tumor microenvironment could contribute to angiogenesis, metastasis, tumor growth and progression(36). So, this pro-inflammation signature induced by Fn infection might accelerate the proliferation of ESCC cells.

Moreover, the cancer related signaling pathway and ESCC progression associated genes were changed significantly after Fn infection, such as the increased expression of DKK1, HAS3, CCND1 and decreased expression of NFE2L2. The overexpression of Dkk1 (Dickkopf1) has been demonstrated can stimulate ESCC cell proliferation independent of Wnt signaling pathway(27). The HAS3 is a member of the hyaluronan synthase family, the products of HAS3 is responsible for secreting the growing hyaluronan polymer into the extracellular. As reported, HAS3 is overexpressed in ESCC clinical samples, and tumor growth can be inhibited by HAS3 knock down in ESCC cells(28). Similarly, the CCND1 and NFE2L2 were also demonstrated can influence the progression of ESCC(6). Taken together, those changes may also contribute to the proliferation of ESCC cells after Fn infection.

It has been reported that Fn can function by activating the TLR4 and MYD88 pathway in CRC cells(22), but in our study the expression levels of TLR4 and MYD88 were not changed (data not shown). Thus, Fn might work through other pathways in ESCC cells, which need to be explored in the following project.

What's more, Dr. Cai et al. reported that breast tumor-resident intracellular microbiota can reorganize the actin cytoskeleton and then enhance the resistance to fluid shear stress of tumor cells during metastasis (13). Interestingly, in our data the DEGs were significantly enriched in the fluid shear stress pathway by KEGG pathway analysis, indicating Fn infection might play roles in the tumor cell metastasis. Meanwhile, literature reported viable Fn can be separated from liver metastases of CRC(20), indicating Fn may migrate with cancer cells to the metastatic site. Thus,

more concerns need to be paid on whether Fn infection can impact the metastasis procedure of ESCC cells by in vivo experiments.

In summary, in the present study we demonstrated the Fn can invade into ESCC cells and promote the proliferation of ESCC cells along with elevated expression of cytokines (CXCL1, IL-6 and IL-1 β) and disturbed expression of some ESCC progression associated genes (DKK1, HAS3, NFE2L2 and CCND1).

Several limitations should be noted here. Firstly, only cell experiments were carried out in the present study. Due to the tumor microenvironment is a complicated system, the effect of *Fusobacterium nucleatum* infection should be studied more thoroughly by in vivo study. Secondly, as for the immune system consist of an important part in tumor formation, progression and therapy, the interaction between Fn and the immune system should be considered.

Supplemental legend

Movie S1. Proliferation of KYSE70 without Fn infection.

Movie S2. Proliferation of KYSE70 with Fn infection.

Materials and methods

Cell culture Human ESCC cell lines KYSE70 and TE5 were cultured in RPMI-1640 medium supplemented with 10% fetal bovine serum (FBS) and 1% Penicillin-Streptomycin Solution (PS). Cells were incubated in the cell incubators at 37°C and 5% CO₂ condition.

Bacterium culture *Fusobacterium nucleatum* (ATCC25586) was obtained from Guangdong Microbial Culture Collection Center (GDMCC) and incubated in Tryptic Soy Broth at 37°C under anaerobic condition. *Bacteroides acidifaciens* (DSM 15896) was obtained from DSMZ-German Collection of Microorganisms and Cell Cultures GmbH and incubated in Brain Heart Infusion Broth at 37°C under anaerobic condition.

Bacterium invasion KYSE70 and TE5 ESCC cells were seeded in a 96-well polypropylene microplates with complete cell culture medium (1640 medium with 10% FBS and 1% PS). Before Fn invasion, cells were washed twice with PBS solution and antibiotic-free medium were added. Fn was collected during logarithmic phase and washed with PBS twice and added to the cells at a MOI400 and were incubated in the cell incubator for 24 hours. After infection, cells were washed with PBS twice and fresh complete medium was added and incubated for the following experiments.

Cell proliferation detection The proliferation of ESCC cell lines was measured by CCK8 assay. Briefly, ESCC cells with or without Fn invasion were washed with PBS before detection. Then 100ul of fresh cell culture medium containing 10% CCK8 solution were added, and kept in the incubator for 1 hr. Then cell viability was determined by measuring the absorbance at a wavelength of 450nm.

Colony formation assay 600 cells were seeded in 60mm Petri Dish (Corning) containing complete RPMI1640 on day 1. The medium was replaced with PS-free cell

culture medium on day 2, then Fn or PBS were added into cell plate at MOI400 and kept in the cell incubator for 24h. The medium was replaced with complete RPMI1640 medium on day 3 and maintained for another 8 days. On day 11, cells were fixed with methanol for 10 mins and stained with 1% crystal violet. After washing with running water, the cell colonies with more than 1mm diameter were counted.

Living cell imaging 2×10^5 KYSE70 cells were seeded in 6-well plates 24 hrs in advance, and the logarithmic phase Fn were added into the plates at MOI400. Then the co-culture system was photographed by BioTek Cytation 1 for more than 48hrs.

Real-Time PCR Real-time PCR was performed for measuring expression levels of differential expressed genes of interest in ESCC cell lines. In brief, the extracted qualified total RNA was reversely transcribed into cDNA with commercialized kits (Vazyme, China), and then the prepared cDNA samples were subjected to real-time PCR reaction using SYBR Green reagent (Vazyme, China) according to the manufacturer's protocol. Primer sequences of targeted genes are listed in Table 1. All reactions were performed in triplicate with an Applied Biosystems StepOnePlus system. Data were normalized to PGK1 expression ($\Delta\Delta CT$ analysis).

Table 1. Real-time PCR primer sequences.

Genes	Forward sequence (5'-3')	Reverse sequence (5'-3')
CXCL1	AAGAACATCCAAAGTGTGAACG	CACTGTTTCAGCATCTTTTCGAT
IL-1 β	GCCAGTGAAATGATGGCTTATT	AGGAGCACTTCATCTGTTTAGG
IL-6	CACTGGTCTTTTGGAGTTTGAG	GGACTTTTGTACTCATCTGCAC
DKK1	TACCAGACCATTGACAACCTACC	TCCATTTTTGCAGTAATTCCCG
HAS3	TTATACAGCTTTTCTACCGGGG	CAGAAGGCTGGACATATAGAGG
CCND1	GTCCTACTTCAAATGTGTGCAG	GGGATGGTCTCCTTCATCTTAG
PGK1	GAGATGATTATTGGTGGTGGAA	AGTCAACAGGCAAGGTAATC

Statistical Analysis Data are expressed as mean \pm SEM. All the experiments (except the living cell imaging) were repeated three times independently. Statistical

comparison between two groups were analyzed by unpaired t-test by GraphPad Prism 7.0. Statistical comparison conducted in more than two groups were analyzed by Two-way ANOVA. $p < 0.05$ was considered to be statistically significant.

Author's contribution

ND, HL, YW and YW performed experiments and data analyzing. ND wrote the manuscript. CC and HC revised the manuscript. WZ and YC designed and guided this project. All authors reviewed and approved the final manuscript.

Acknowledgement This work was supported by the fund of Guangdong Basic and Applied Basic Research Foundation (2019B030302012), Major Program of Shenzhen Bay Laboratory (S201101004), Shenzhen Key Project of Science and Technology (JCYJ20200109120425045), Shenzhen Bay Laboratory Open Program (SZBL2020090501003), the National Natural Science Foundation of China (U21A20372, 81972613, 82103143), the China National Postdoctoral Program for Innovative Talents (BX2021194), the National Key R&D Program of China (2021YFC2501001), “San-ming” Project of Medicine in Shenzhen (SZSM201812088), Natural Science Foundation of Guangdong Province (2020A1515010431).

References

1. Bray F, Ferlay J, Soerjomataram I, Siegel RL, Torre LA, Jemal A. 2018. Global cancer statistics 2018: GLOBOCAN estimates of incidence and mortality worldwide for 36 cancers in 185 countries. *CA Cancer J Clin* 68:394-424.
2. Zheng R, Zhang S, Zeng H, Wang S, Sun K, Chen R, Li L, Wei W, He J. 2022. Cancer incidence and mortality in China, 2016. *Journal of the National Cancer Center* 2:1-9.
3. Lin Y, Totsuka Y, Shan B, Wang C, Wei W, Qiao Y, Kikuchi S, Inoue M, Tanaka H, He Y. 2017. Esophageal cancer in high-risk areas of China: research progress and challenges. *Ann Epidemiol* 27:215-221.
4. Pennathur A, Gibson MK, Jobe BA, Luketich JD. 2013. Oesophageal carcinoma. *Lancet* 381:400-12.
5. Abnet CC, Arnold M, Wei W-Q. 2018. Epidemiology of Esophageal Squamous Cell Carcinoma. *Gastroenterology* 154:360-373.
6. Cui Y, Chen H, Xi R, Cui H, Zhao Y, Xu E, Yan T, Lu X, Huang F, Kong P, Li Y, Zhu X, Wang J, Zhu W, Wang J, Ma Y, Zhou Y, Guo S, Zhang L, Liu Y, Wang B, Xi Y, Sun R, Yu X, Zhai Y, Wang F, Yang J, Yang B, Cheng C, Liu J, Song B, Li H, Wang Y, Zhang Y, Cheng X, Zhan Q, Li Y, Liu Z. 2020. Whole-genome sequencing of 508 patients identifies key molecular features associated with poor prognosis in esophageal squamous cell carcinoma. *Cell Res* doi:10.1038/s41422-020-0333-6.
7. Spor A, Koren O, Ley R. 2011. Unravelling the effects of the environment and host genotype on the gut microbiome. *Nat Rev Microbiol* 9:279-90.

8. Li Q, Hu W, Liu WX, Zhao LY, Huang D, Liu XD, Chan H, Zhang Y, Zeng JD, Coker OO, Kang W, Man Ng SS, Zhang L, Wong SH, Gin T, Vai Chan MT, Wu JL, Yu J, Kei Wu WK. 2020. *Streptococcus thermophilus* inhibits colorectal tumorigenesis through secreting beta-galactosidase. *Gastroenterology* doi:10.1053/j.gastro.2020.09.003.
9. Montalban-Arques A, Katkeviciute E, Busenhardt P, Bircher A, Wirbel J, Zeller G, Morsy Y, Borsig L, Glaus Garzon JF, Muller A, Arnold IC, Artola-Boran M, Krauthammer M, Sintsova A, Zamboni N, Leventhal GE, Berchtold L, de Wouters T, Rogler G, Baebler K, Schwarzfischer M, Hering L, Olivares-Rivas I, Atrott K, Gottier C, Lang S, Boyman O, Fritsch R, Manz MG, Spalinger MR, Scharl M. 2021. Commensal Clostridiales strains mediate effective anti-cancer immune response against solid tumors. *Cell Host Microbe* doi:10.1016/j.chom.2021.08.001.
10. Liu Y, Fu K, Wier EM, Lei Y, Hodgson A, Xu D, Xia X, Zheng D, Ding H, Sears CL, Yang J, Wan F. 2022. Bacterial Genotoxin Accelerates Transient Infection-Driven Murine Colon Tumorigenesis. *Cancer Discov* 12:236-249.
11. Zhang Y, Zhang L, Zheng S, Li M, Xu C, Jia D, Qi Y, Hou T, Wang L, Wang B, Li A, Chen S, Si J, Zhuo W. 2022. *Fusobacterium nucleatum* promotes colorectal cancer cells adhesion to endothelial cells and facilitates extravasation and metastasis by inducing ALPK1/NF-kappaB/ICAM1 axis. *Gut Microbes* 14:2038852.
12. Bertocchi A, Carloni S, Ravenda PS, Bertalot G, Spadoni I, Lo Cascio A, Gandini S, Lizier M, Braga D, Asnicar F, Segata N, Klaver C, Brescia P, Rossi E, Anselmo A, Guglietta S, Maroli A, Spaggiari P, Tarazona N, Cervantes A, Marsoni S, Lazzari L, Jodice MG, Luise C, Erreni M, Pece S, Di Fiore PP, Viale G, Spinelli A, Pozzi C,

- Penna G, Rescigno M. 2021. Gut vascular barrier impairment leads to intestinal bacteria dissemination and colorectal cancer metastasis to liver. *Cancer Cell* 39:708-724 e11.
13. Fu A, Yao B, Dong T, Chen Y, Yao J, Liu Y, Li H, Bai H, Liu X, Zhang Y, Wang C, Guo Y, Li N, Cai S. 2022. Tumor-resident intracellular microbiota promotes metastatic colonization in breast cancer. *Cell* 185:1356-1372 e26.
 14. Gao Y, Bi D, Xie R, Li M, Guo J, Liu H, Guo X, Fang J, Ding T, Zhu H, Cao Y, Xing M, Zheng J, Xu Q, Xu Q, Wei Q, Qin H. 2021. *Fusobacterium nucleatum* enhances the efficacy of PD-L1 blockade in colorectal cancer. *Signal Transduct Target Ther* 6:398.
 15. Anonymous. 2020-8-13. Microbiome-derived inosine modulates response to checkpoint inhibitor immunotherapy. *Science*.
 16. Yin J, Dong L, Zhao J, Wang H, Li J, Yu A, Chen W, Wei W. 2020. Composition and consistence of the bacterial microbiome in upper, middle and lower esophagus before and after Lugol's iodine staining in the esophagus cancer screening. *Scand J Gastroenterol* 55:1467-1474.
 17. Deshpande NP, Riordan SM, Castano-Rodriguez N, Wilkins MR, Kaakoush NO. 2018. Signatures within the esophageal microbiome are associated with host genetics, age, and disease. *Microbiome* 6:227.
 18. Brennan CA, Garrett WS. 2019. *Fusobacterium nucleatum* - symbiont, opportunist and oncobacterium. *Nat Rev Microbiol* 17:156-166.
 19. Hong J, Guo F, Lu SY, Shen C, Ma D, Zhang X, Xie Y, Yan T, Yu T, Sun T, Qian Y, Zhong M, Chen J, Peng Y, Wang C, Zhou X, Liu J, Liu Q, Ma X, Chen YX, Chen H,

- Fang JY. 2020. *F. nucleatum* targets lncRNA ENO1-IT1 to promote glycolysis and oncogenesis in colorectal cancer. *Gut* doi:10.1136/gutjnl-2020-322780.
20. Bullman S, Pedamallu CS, Sicinska E, Clancy TE, Zhang X, Cai D, Neuberger D, Huang K, Guevara F, Nelson T, Chipashvili O, Hagan T, Walker M, Ramachandran A, Diosdado B, Serna G, Mulet N, Landolfi S, Ramon YCS, Fasani R, Aguirre AJ, Ng K, Elez E, Ogino S, Taberner J, Fuchs CS, Hahn WC, Nuciforo P, Meyerson M. 2017. Analysis of *Fusobacterium* persistence and antibiotic response in colorectal cancer. *Science* 358:1443-1448.
21. Guo P, Tian Z, Kong X, Yang L, Shan X, Dong B, Ding X, Jing X, Jiang C, Jiang N, Yu Y. 2020. FadA promotes DNA damage and progression of *Fusobacterium nucleatum*-induced colorectal cancer through up-regulation of chk2. *J Exp Clin Cancer Res* 39:202.
22. Yu T, Guo F, Yu Y, Sun T, Ma D, Han J, Qian Y, Kryczek I, Sun D, Nagarsheth N, Chen Y, Chen H, Hong J, Zou W, Fang JY. 2017. *Fusobacterium nucleatum* Promotes Chemoresistance to Colorectal Cancer by Modulating Autophagy. *Cell* 170:548-563.e16.
23. Yamamura K, Baba Y, Nakagawa S, Mima K, Miyake K, Nakamura K, Sawayama H, Kinoshita K, Ishimoto T, Iwatsuki M, Sakamoto Y, Yamashita Y, Yoshida N, Watanabe M, Baba H. 2016. Human Microbiome *Fusobacterium Nucleatum* in Esophageal Cancer Tissue Is Associated with Prognosis. *Clin Cancer Res* 22:5574-5581.
24. Yamamura K, Izumi D, Kandimalla R, Sonohara F, Baba Y, Yoshida N, Koder Y,

- Baba H, Goel A. 2019. Intratumoral *Fusobacterium Nucleatum* Levels Predict Therapeutic Response to Neoadjuvant Chemotherapy in Esophageal Squamous Cell Carcinoma. *Clin Cancer Res* 25:6170-6179.
25. Liu Y, Baba Y, Ishimoto T, Tsutsuki H, Zhang T, Nomoto D, Okadome K, Yamamura K, Harada K, Eto K, Hiyoshi Y, Iwatsuki M, Nagai Y, Iwagami S, Miyamoto Y, Yoshida N, Komohara Y, Ohmuraya M, Wang X, Ajani JA, Sawa T, Baba H. 2020. *Fusobacterium nucleatum* confers chemoresistance by modulating autophagy in oesophageal squamous cell carcinoma. *British Journal of Cancer* doi:10.1038/s41416-020-01198-5.
26. Zhang N, Liu Y, Yang H, Liang M, Wang X, Wang M, Kong J, Yuan X, Zhou F. 2021. Clinical Significance of *Fusobacterium nucleatum* Infection and Regulatory T Cell Enrichment in Esophageal Squamous Cell Carcinoma. *Pathol Oncol Res* 27:1609846.
27. Kimura H, Sada R, Takada N, Harada A, Doki Y, Eguchi H, Yamamoto H, Kikuchi A. 2021. The Dickkopf1 and FOXM1 positive feedback loop promotes tumor growth in pancreatic and esophageal cancers. *Oncogene* 40:4486-4502.
28. Twarock S, Freudenberger T, Poscher E, Dai G, Jannasch K, Dullin C, Alves F, Prenzel K, Knoefel WT, Stoecklein NH, Savani RC, Homey B, Fischer JW. 2011. Inhibition of oesophageal squamous cell carcinoma progression by in vivo targeting of hyaluronan synthesis. *Mol Cancer* 10:30.
29. Bi Y, Guo S, Xu X, Kong P, Cui H, Yan T, Ma Y, Cheng Y, Chen Y, Liu X, Zhang L, Cheng C, Xu E, Qian Y, Yang J, Song B, Li H, Wang F, Hu X, Liu X, Niu X, Zhai Y, Liu J, Li Y, Cheng X, Cui Y. 2020. Decreased ZNF750 promotes angiogenesis in a

paracrine manner via activating DANCR/miR-4707-3p/FOXC2 axis in esophageal squamous cell carcinoma. *Cell Death Dis* 11:296.

30. Cheng C, Zhou Y, Li H, Xiong T, Li S, Bi Y, Kong P, Wang F, Cui H, Li Y, Fang X, Yan T, Li Y, Wang J, Yang B, Zhang L, Jia Z, Song B, Hu X, Yang J, Qiu H, Zhang G, Liu J, Xu E, Shi R, Zhang Y, Liu H, He C, Zhao Z, Qian Y, Rong R, Han Z, Zhang Y, Luo W, Wang J, Peng S, Yang X, Li X, Li L, Fang H, Liu X, Ma L, Chen Y, Guo S, Chen X, Xi Y, Li G, Liang J, Yang X, Guo J, et al. 2016. Whole-Genome Sequencing Reveals Diverse Models of Structural Variations in Esophageal Squamous Cell Carcinoma. *Am J Hum Genet* 98:256-74.
31. Yan T, Cui H, Zhou Y, Yang B, Kong P, Zhang Y, Liu Y, Wang B, Cheng Y, Li J, Guo S, Xu E, Liu H, Cheng C, Zhang L, Chen L, Zhuang X, Qian Y, Yang J, Ma Y, Li H, Wang F, Liu J, Liu X, Su D, Wang Y, Sun R, Guo S, Li Y, Cheng X, Liu Z, Zhan Q, Cui Y. 2019. Multi-region sequencing unveils novel actionable targets and spatial heterogeneity in esophageal squamous cell carcinoma. *Nat Commun* 10:1670.
32. Koenig JE, Spor A, Scalfone N, Fricker AD, Stombaugh J, Knight R, Angenent LT, Ley RE. 2011. Succession of microbial consortia in the developing infant gut microbiome. *Proc Natl Acad Sci U S A* 108 Suppl 1:4578-85.
33. He Y, Wu W, Wu S, Zheng HM, Li P, Sheng HF, Chen MX, Chen ZH, Ji GY, Zheng ZD, Mujagond P, Chen XJ, Rong ZH, Chen P, Lyu LY, Wang X, Xu JB, Wu CB, Yu N, Xu YJ, Yin J, Raes J, Ma WJ, Zhou HW. 2018. Linking gut microbiota, metabolic syndrome and economic status based on a population-level analysis. *Microbiome* 6:172.

34. Parida S, Wu S, Siddharth S, Wang G, Muniraj N, Nagalingam A, Hum C, Mistriotis P, Hao H, Talbot CC, Konstantopoulos K, Gabrielson KL, Sears CL, Sharma D. 2021. A pro-carcinogenic colon microbe promotes breast tumorigenesis and metastatic progression and concomitantly activates Notch and betacatenin axes. *Cancer Discov* doi:10.1158/2159-8290.CD-20-0537.
35. Anonymous. 2020-5. The human tumor microbiome is composed of tumor type-specific intracellular bacteria. *Science*.
36. Lazenec G, Richmond A. 2010. Chemokines and chemokine receptors: new insights into cancer-related inflammation. *Trends Mol Med* 16:133-44.

Review comments

Fusobacterium nucleatum is a cancer-related bacterium. A large number of studies have revealed the relationship between Fn and cancers from different angles since 2012. However, those researches mainly focus on colorectal cancer. For the relationship of Fn and esophageal cancer, the Japanese first reported in *Clin Cancer Res* in 2016 that high load of Fn was positively correlated with the short survival period of esophageal cancer. In 2019, this team found that Fn reduced the effect of neoadjuvant chemotherapy. In 2022, they observed with transmission electron microscope that Fn could invade ESCC cells and affect gene and protein expression, and found the promotion of NOD1 receptor - RIPK2 -NF- κ B in the invasion, migration and proliferation of cancer cells. In addition, in 2021, through the analysis of 41 Chinese patients with esophageal cancer, Zhen Li et al found that the Fn was significantly related to the pathological stage and clinical stage of esophageal cancer, and also to the metastasis of tumor.

In order to explore the mechanism of Fn on esophageal cancer, the ESCC cell line and Fn were co-cultured in this study. Two conclusions were obtained:

1. Fn can invade ESCC cells, thus enhancing the proliferation of ESCC cells.
2. After Fn infection, gene expression in ESCC cells was changed, and gene expression related to inflammation related signals and cancer signals was up-regulated.

There are few studies on the correlation between Fn and esophageal cancer, and its corresponding mechanism is not clear. The topic of this study is very meaningful, and the language description of the manuscript is clear. However, the mechanism of Fn and the proliferation of esophageal cancer has not been clearly proposed. It is suggested to supplement the corresponding data to better understand the pro-proliferative effect of Fn in esophageal cancer.

The suggested supplementary data are as follows:

1. Why is there a significant difference in the morphology of Fn in Fig. 1? How

can you prove that the marker is a bacterium, not a cell structure or other components?

2. In Fig2b, Does *Bacteroides acidifaciens* also enter the cell?

3. In this study, there is only co-culture model in vitro, and corresponding animal experiments should be supplemented to observe whether Fn promotes the proliferation of esophageal cancer in vivo;

4. To clarify the specific target and corresponding mechanism of Fn promoting the proliferation of esophageal cancer;

Dear Dr. Emad El-Omar,

I hope this letter finds everything going well with you.

Thank you very much for your letter dated September 4th, 2022, and the referees' reports. Based on your comment and request, we have made extensive modification on the original manuscript entitled "*Fusobacterium nucleatum* infection induces malignant proliferation of esophageal squamous cell carcinoma cells by putrescine production" (previous manuscript ID #Spectrum02759-22)". Here, we attached revised manuscript, in the format of MS word, for your approval. A document answering every question from the referees was also summarized and enclosed. A revised manuscript with the correction sections red marked was attached as the supplemental material and for easy check/editing purpose. Hope these will make it more suitable for publication in *Microbiology Spectrum*. Should you have any questions, please contact us without hesitate.

With my best regard!

Sincerely,

Yongping Cui, Ph.D.

Dear reviewers,

We are very grateful to your encouraging and thoughtful comments and suggestions regarding our original submission. In response to these comments, we have made a number of modifications to our manuscript and performed additional suggested experiments. Below we detail these specific modifications with the specific comments from the reviewers followed by our response. We hope the reviewers will find this manuscript improved following these changes and more suitable for publication in *Microbiology Spectrum*.

Reviewer #1 (Comments for the Author):

Fusobacterium nucleatum is a cancer-related bacterium. A large number of studies have revealed the relationship between Fn and cancers from different angles since 2012. However, those researches mainly focus on colorectal cancer. For the relationship of Fn and esophageal cancer, the Japanese first reported in *Clin Cancer Res* in 2016 that high load of Fn was positively correlated with the short survival period of esophageal cancer. In 2019, this team found that Fn reduced the effect of neoadjuvant chemotherapy. In 2022, they observed with transmission electron microscope that Fn could invade ESCC cells and affect gene and protein expression, and found the promotion of NOD1 receptor - RIPK2 -NF- κ B in the invasion, migration and proliferation of cancer cells. In addition, in 2021, through the analysis of 41 Chinese patients with esophageal cancer, Zhen Li et al found that the Fn was significantly related to the pathological stage and clinical stage of esophageal cancer, and also to the metastasis of tumor.

In order to explore the mechanism of Fn on esophageal cancer, the ESCC cell line and Fn were co-cultured in this study. Two conclusions were obtained: 1. Fn can invade ESCC cells, thus enhancing the proliferation of ESCC cells. 2. After Fn infection, gene expression in ESCC cells was changed, and gene expression related to inflammation related signals and cancer signals was up regulated.

There are few studies on the correlation between Fn and esophageal cancer, and its corresponding mechanism is not clear. The topic of this study is very meaningful, and the language description of the manuscript is clear. However, the mechanism of Fn and the proliferation of esophageal cancer has not been clearly proposed. It is suggested to supplement the corresponding data to better understand the pro-proliferative effect of Fn in esophageal cancer.

The suggested supplementary data are as follows:

Q1. Why is there a significant difference in the morphology of Fn in Fig. 1? How can you prove that the marker is a bacterium, not a cell structure or other components?

Response 1: Indeed, we can see different morphology of Fn in Fig.1 (Fig. 3A), because cell is a three-dimensional structure, the bacteria were random distribution in the cells, and the picture were photographed from one layer of the cell, thus we can see different morphology. And they are identified as Fn bacteria, because we can see the “cell wall” in light gray at the margin of those bacteria (Fig. S1). In order to better clarify that Fn can invade into ESCC cells, the classic “differential immunofluorescence experiment” were carried out (Mauro Castellarin, et al., Genome Research, 2012; Poonam Dharmani, et al., Infection and Immunity, 2011; J Pizarro-Cerdá, et al., Methods in Microbiology, 2022), the Intracellular-Fn were marked as green in the merge picture (Fig. 3B).

Fig. S1 The “cell wall” in light gray at the margin of those bacteria. The red arrows indicate the cell wall structure of bacteria.

Fig. 3B. Intracellular Fn was detect by differential immunofluorescence experiment. Red, Fn outside of ESCC cells; Green, Fn both inside and outside ESCC cells. The Fn labeled in green in the Merge picture were intracellular. Scale bar = 10µm.

Q2. In Fig2b, Does *Bacteroides acidifaciens* also enter the cell?

Response 2: The cells were collected after co-culture with *Bacteroides acidifaciens* (Ba) for 24h, and the TEM result presented that Ba can’t entry into ESCC cells. (Fig. S2).

Fig. S2. *Bacteroides acidifaciens* (BA) cannot invasion into ESCC cells. BA-infected cells were detected by transmission electron microscopy. Scale bar = 20 μ m.

Q3. In this study, there is only co-culture model in vitro, and corresponding animal experiments should be supplemented to observe whether Fn promotes the proliferation of esophageal cancer in vivo;

Response 3: To be honest, I quite agree with you that the in vivo experiments should be done to better understand the influence of Fn infection during tumor progression and metastasis. Unfortunately, due to Fn is a pathogenic bacterium, we have consulted 3 available animal experimental centers, Fn relative experiment were not permitted. And this limitation was discussed in the final part of Discussion. Hope you will understand this problem.

Q4. To clarify the specific target and corresponding mechanism of Fn promoting the proliferation of esophageal cancer;

Response 4: As reported, Fn can induced tumor progression or metastasis in several cancer types by multiple mechanisms. Such as, Fn can induce cytokines secreting (GM-CSF, CXCL1, IL-8 and MIP-3 α) in pancreatic cancer (Barath Udayasuryan, et al., Science Signaling, 2022). And Fn can activate YAP signaling and inhibiting the expression of FOXD3, subsequently reduced METTL3 transcription and promoted the KIF26B expression by reducing its m⁶A levels in CRC (Shujie Chen, et al., Nature Communications, 2022). Even, Fn can promote the CRC cell metastasis by changing the exosomes contents (Songhe Guo, et al., Gut, 2020).

In our study, we found that the SAT1 were highly expressed after Fn infection (Fig. 4A,) and the relative expression level of SAT1 were increased gradually with time extension after Fn invasion (Fig. 4B), indicating the SAT1 may take part in the Fn-induced cell proliferation. As the SAT1 encodes an rate-limiting enzyme in the

polyamine metabolism pathway (Fig. 4C), thus the polyamine concentrations after Fn infection were detected. The putrescine, spermidine, spermine were increased significantly after Fn infection (Fig. 4D). Those polyamines were crucial for cancer cell proliferation and were targets of cancer therapy (Raboer A. et al., Nature Review Cancer, 2018; Cassandra E. Holbert, et al., Nature Review Cancer, 2022). Among the elevated polyamines, the putrescine increased most significantly with more than 6-fold (Fig. 4D). And by analyzing polyamines concentrations in Fn microbes itself, we found that Fn itself can produce extremely high levels of putrescine than other polyamines (Fig.4 E). At the meantime the expression level of Ornithine Decarboxylase (ODC), which encodes the enzyme involved in putrescine synthesis, were stable in 6 hrs after Fn infection and decreased significantly at the 24hrs after Fn infection (Fig. 4F), indicated the synthesis of putrescine in ESCC cells were not active after Fn infection. Taken together, the dramatic increased putrescine in Fn-infected-ESCC cells were produced by Fn. In order to clarify the influence of elevated putrescine induced by Fn infection, we mimic the increased cell putrescine level by added putrescine into cell culture medium and found the putrescine can increase cell proliferation in a dose dependent manner (Fig. 4G) Taken together, we believe the disturbed polyamine metabolism by Fn infection in an important factor which influence the proliferation of ESCC cells.

Fig 4. Fn can produce putrescine disturbed the polyamines metabolism pathway in ESCC cells and promote ESCC cell proliferation. A. Volcano plot showing the changes of genes (fold change ≥ 2) in KYSE70 after Fn infection. SAT1 was increased significantly after Fn infection. B. Relative expression level of SAT1 increased after Fn infection with time extension. C. Polyamine metabolism map. Ornithine decarboxylase (ODC) is a rate limiting step in putrescine production. The spermidine/spermine N¹-acetyltransferase 1 (SAT1) can acetylate the N¹ position of spermidine or spermine, allowing either export or oxidative back-conversion by peroxisomal acetyl polyamine oxidase (PAO). D. The concentration of polyamines in ESCC cells with/without Fn infection for 24hrs. E. The polyamine concentration in Fn

*bacterium. F. Relative expression level of ODC keep stable in the 6hrs after Fn infection and decreased sharply at the 24hr of Fn infection. G. CCK8 assay was used to examine the cell viability of KYSE70 by increased concentration of putrescine. * $p < 0.05$, ** $p < 0.01$, *** $p < 0.001$, **** $p < 0.0001$.*

Reviewer #2 (Comments for the Author):

Most of the information provided in this study were not arranged sequentially. The author needs to follow standard procedure so that the study will be easier to understand.

Please note my input below:

Q1. Topic - I am suggesting a new title: Induction of Malignant Proliferation of Esophageal Squamous Cell Carcinoma by *Fusobacterium nucleatum*

Response 1: Thank you for your advice, due to some results have been added into the manuscript, the topic was changed into: *Fusobacterium nucleatum* Infection Induces Malignant Proliferation of Esophageal Squamous Cell Carcinoma Cells by Putrescine Production. If you have any suggestions for improvement, please tell me!

Q2. Abstract - You need to be focused on your topic. Give a concise summary of the background of the study, objective, methods used for the study, result obtained (findings) and conclusion (in line with the objective). Please use this concept to re-write your abstract. Presently everything is muddled up.

Response 2: Abstract has been modified, thank you for your kindly advise.

Q3. Introduction - You started well but veered off. Some introduction information were stated under discussion without sufficient references

Response 3: Instruction has been modified.

Q4. Method - It will carry more weight and portray a better flow if it comes before discussion, please relocate.

Response 4: Due to methods were arranged after the discussion in the published articles in Microbiology Spectrum, so we didn't replace the Methods part. Please forgive that brings you the inconvenience.

Q5. Result - You veered off. The headings were not concise. Please report what you found out

Response 5: The headings have been revised!

Reviewer #3 (Comments for the Author):

Through a series of in vitro experiments, Ding and colleagues found that Fn could invade ESCC cells and further enhance the proliferation ability of ESCC cells. In addition, the expression levels of CXCL1, IL-6 and IL-1 β in ESCC cells were significantly increased after Fn infection. I hope the authors may find some of the following suggestions useful:

Major points:

Q1. There is no conclusion description in the abstract.

Response 1: Thanks for your advice, the conclusion has been added

Q2. There is no validation and molecular biological rescue experiments for RNA-seq results regulated by Fn.

Response 2: In the newly revised manuscript, we found that the SAT1 were highly expressed after Fn infection (Fig. 4A), and the relative expression level of SAT1 were increased gradually with time extension after Fn invasion (Fig. 4B) indicating the SAT1 may take part in the Fn-induced cell proliferation. As the SAT1 encodes an rate-limiting enzyme in the polyamine metabolism pathway (Fig. 4C), thus the polyamine levels after Fn infection were detected, and the putrescine, spermidine, spermine were increased significantly after Fn infection (Fig. 4D). And all these polyamines were crucial for cancer cell proliferation and were the target of cancer therapy (Raboer A. et al., Nature Review Cancer, 2018; Cassandra E. Holbert, et al., Nature Review Cancer, 2022). Among the elevated polyamines, the putrescine increased most significantly with more than 6-fold (Fig. 4D). And by analyzing polyamines concentrations in Fn microbes itself, we found that Fn itself can product extremely high levels of putrescine than other polyamines (Fig.4 E). At the meantime the expression level of Ornithine Decarboxylase (ODC) were stable in 6 hrs after Fn infection and decreased significantly at the 24hrs after Fn infection (Fig. 4F). The ODC gene encode the enzyme involved in putrescine synthesis, and the decreased expression level of ODC indicated the synthesis of putrescine in ESCC cells were not active after Fn infection. Taken together, the dramatic increased putrescine in Fn-infected-ESCC cells were produced by Fn. In order to clarify the influence of elevated putrescine induced by Fn infection, we mimic the increased cell putrescine level by added putrescine into cell culture medium and found the putrescine can increase cell proliferation in a dose dependent manner (Fig. 4G) Taken together, we believe the disturbed polyamine metabolism by Fn infection in an important factor which influence the proliferation of ESCC cells.

Fig 4. F*n* can produce putrescine disturbed the polyamines metabolism pathway in ESCC cells and promote ESCC cell proliferation. A. Volcano plot showing the changes of genes (fold change ≥ 2) in KYSE70 after F*n* infection. SAT1 was increased significantly after F*n* infection. B. Relative expression level of SAT1 increased after F*n* infection with time extension. C. Polyamine metabolism map. Ornithine decarboxylase (ODC) is a rate limiting step in putrescine production. The spermidine/spermine N¹-acetyltransferase 1 (SAT1) can acetylate the N¹ position of spermidine or spermine, allowing either export or oxidative back-conversion by peroxisomal acetylpolyamine oxidase (PAO). D. The concentration of polyamines in ESCC cells with/without F*n* infection for 24hrs. E. The polyamine concentration in F*n* bacterium. F. Relative expression level of ODC keep stable in the 6hrs after F*n* infection and decreased sharply at the 24hr of F*n* infection. G. CCK8 assay was used to examine the cell viability of KYSE70 by increased concentration of putrescine. * $p < 0.05$, ** $p < 0.01$, *** $p < 0.001$, **** $p < 0.0001$.

Q3. What is the specific pathway of F*n* invasion into cells? It should be discussed or validated accordingly

Response 3: Some studies have demonstrated that F*n* can binding to and 11 amino acid region of E-cadherin by its FadA virulence factor and induced CRC cell growth (Rubinstein MR, et al., Cell Host Microbe, 2013). Otherwise, the Fap2 of F*n* also can binding to the D-galactose- β (1-3)-N-acetyl-D-galactosamine (Gal-GalNAc) of cancer cells in CRC and breast tumor cells(Abed J, et al., Cell Host Microbe, 2016; Parhi L, et al., Nature Communications, 2020). Now, it has been discussed in the Discussion.

Q4. For the change of tumor cell function caused by F*n*, it is not enough to only observe the proliferation ability. Does it affect the invasion, migration and apoptosis of cells?

Response 4: Thanks for your good question! The Fn infection can significantly promoted ESCC cell migration after 6 hrs infection (Fig. 2C). However, the KYSE70 ESCC cells didn't invasion in the transwell assay covered with Matrigel, so we don't know whether Fn infection can promote ESCC cell invasion. As for the apoptosis, the ESCC cells infected with/without Fn were collected after 6hrs infection, there was no difference in the apoptotic cell rates (Fig. S3).

Fig. 2C. Fn infection can promote the migration of ESCC cells. The migration ability of KYSE70 with/without Fn infection was examined by Transwell experiment. The migrated cells from 3 independent experiments were counted and analyzed. ***, $p < 0.001$.

Fig. S3. Fn infection didn't impact the apoptosis of ESCC cells. The apoptosis of ESCC cells with/without Fn infection were collected and detected by flow cytometry. The rates of early apoptosis and late apoptosis cells were analyzed (right).

Q5. The abundance of Fn in clinical patients should be supplemented. Is there a correlation between Fn level and clinicopathological parameters?

Response 5: To answer this question, we collected 73 ESCC samples with paired tumor and adjacent normal tissues from Shanxi province, China. We found Fn were more abundant in tumor tissues and were significantly correlated with shorter survival (New Fig 1A and B).

Fig. 1A and B. The relative abundance of Fn were higher in Tumor than adjacent normal tissues and were correlated with worse prognosis in clinical ESCC samples from China. **A.** The relative abundance of Fn in clinical ESCC samples were detected by TaqMan qPCR analysis. **B.** The high abundance of Fn infiltration in tumor tissues were significantly correlated with shorter survival. ** $p < 0.01$.

January 9, 2023

Dr. Yongping Cui

Shenzhen Key Laboratory of Gastrointestinal Cancer Translational Research, Peking University Shenzhen Hospital, Shenzhen Peking University-the Hong Kong University of Science and Technology (PKU-HKUST) Medical Center. Institute of Cancer Research, Shenzhen Bay Laboratory

Shenzhen Key Laboratory of Gastrointestinal Cancer Translational Research, Peking University Shenzhen Hospital, Shenzhen Peking University-the Hong Kong University of Science and Technology (PKU-HKU

Shenzhen

China

Re: Spectrum02759-22R1 (*Fusobacterium nucleatum* infection induces malignant proliferation of esophageal squamous cell carcinoma cell by putrescine production)

Dear Dr. Yongping Cui:

Your manuscript has been accepted, and I am forwarding it to the ASM Journals Department for publication. You will be notified when your proofs are ready to be viewed.

Sincerely,

Florence Doucet-Populaire
Editor, Microbiology Spectrum

Journals Department
The author partially answered the question:

1. Why is there a significant difference in the morphology of Fn in Fig. 1? How can you prove that the marker is a bacterium, not a cell structure or other components?

The clear answer through experiments is bacterial structure.

2. My question is "It is recommended to supplement the negative control in Fig2B. to clarify whether Fn itself or its metabolite promotes cell proliferation." Why did it become " In Fig2b, Does *Bacteroides acidifaciens* also enter the cell? " ?

3. In this study, there is only co-culture model in vitro, and corresponding animal experiments should be supplemented to observe whether Fn promotes the proliferation of esophageal cancer in vivo;

There is no answer to this. There have been a lot of literature reports on animal experiments of fn. In vitro experiments cannot completely replace the results in vivo.

4. To clarify the specific target and corresponding mechanism of Fn promoting the proliferation of esophageal cancer;

The article only analyzes the RNA-seq data, and proposes SAT1 were highly expressed after Fn infection, which activates polyamine metabolism path. Why focus on SAT1? Does the protein level of sat1 change? After inhibiting polyamine metabolism pathway, will cells still proliferate? In animal models and clinical specimens, whether Fn also plays a role through this target and corresponding mechanisms.